# Personalised Prevention of Falls in Persons with Dementia—A Registry-Based Study

**DOI:** 10.3390/geriatrics10040106

**Published:** 2025-08-06

**Authors:** Per G. Farup, Knut Hestad, Knut Engedal

**Affiliations:** 1Department of Research and Innovation, Innlandet Hospital Trust, P.O. Box 104, N-2381 Brumunddal, Norway; knut.hestad@inn.no; 2Department of Clinical and Molecular Medicine, Faculty of Medicine and Health Sciences, Norwegian University of Science and Technology, N-7491 Trondheim, Norway; 3The Norwegian National Center for Aging and Health, Vestfold Hospital Trust, N-3103 Tønsberg, Norway; knutengedal@outlook.com; 4Department of Geriatric Medicine, Oslo University Hospital, N-0424 Oslo, Norway

**Keywords:** Alzheimer’s disease, cognitive functions, dementia with Lewy bodies, frailty, frontotemporal dementia, vascular dementia

## Abstract

Background/Objectives: Multifactorial prevention of falls in persons with dementia has minimal or non-significant effects. Personalised prevention is recommended. We have previously shown that gait speed, basic activities of daily living (ADL), and depression (high Cornell scores) were independent predictors of falls in persons with mild and moderate cognitive impairment. This study explored person-specific risks of falls related to physical, mental, and cognitive functions and types of dementia: Alzheimer’s disease (AD), vascular dementia (VD), mixed Alzheimer’s disease/vascular dementia (MixADVD), frontotemporal dementia (FTD), and dementia with Lewy bodies (DLB). Methods: The study used data from “The Norwegian Registry of Persons Assessed for Cognitive Symptoms” (NorCog). Differences between the dementia groups and predictors of falls, gait speed, ADL, and Cornell scores were analysed. Results: Among study participants, 537/1321 (40.7%) reported a fall in the past year, with significant variations between dementia diagnoses. Fall incidence increased with age, comorbidity/polypharmacy, depression, and MAYO fluctuation score and with reduced physical activity, gait speed, and ADL. Persons with VD and MixADVD had high fall incidences and impaired gait speed and ADL. Training of physical fitness, endurance, muscular strength, coordination, and balance and optimising treatment of comorbidities and medication enhance gait speed. Improving ADL necessitates, in addition, relief of cognitive impairment and fluctuations. Relief of depression and fluctuations by psychological and pharmacological interventions is necessary to reduce the high fall risk in persons with DLB. Conclusions: The fall incidence and fall predictors varied significantly. Personalised interventions presuppose knowledge of each individual’s fall risk factors.

## 1. Introduction

The incidence of falls and fall injuries increases with age and in persons with dementia and frailty and results in reduced quality of life, years lived with disability, and increased mortality [1,2,3]. All types of dementia are, to varying degrees, associated with reduced physical, mental, and cognitive functions. In frail older people and people with dementia, the annual fall incidence is nearly double that of cognitively healthy people; annual incidence rates of up to 80% have been reported [3]. In Norway, health spending related to dementia and falls rank at the top, at 10.2% and 4.6% of total expenditure, respectively [4]. Effective prevention of falls could improve people’s quality of life and reduce the community’s health spending.

Prevention of falls has been disappointing. Multifactorial interventions targeting multiple risk factors of falls are ambitious but have had little to no effect [5,6,7]. Peek et al. [7] concluded in a literature review “There is currently insufficient evidence to endorse any intervention to reduce falls for people living with dementia in any setting”. To resign a considerable health problem is no alternative. The latest expert recommendations and world guidelines recommend interventions tailored to each person’s needs [8,9]. Expert recommendations have stated that the lack of emphasis on the importance of tailored interventions, regular monitoring, and psychosocial support is a crucial gap in existing clinical guidelines [9]. Tailored, person-specific interventions presuppose knowledge of the individual’s fall risk factors.

We have previously reported an annual fall incidence of 3774/9525 (39.6%) in a population with impaired cognitive function [10]. Falls were associated with frailty. The type and degree of symptoms indicating frailty varied between persons and across types of dementia. Reduced gait speed, impaired basic ADL, and depression were independent and the main predictors of falls [10]. A detailed and individual mapping of these functions and factors associated with them is a prerequisite for targeted interventions. This follow-up study in persons with mild to moderate dementia aimed to explore the risk of falls related to types of dementia and physical, mental, and cognitive functions available for personalised fall prevention.

## 2. Materials and Methods

### 2.1. Data Materials

From 2009 to 2021, the national research and quality registry “The Norwegian Registry of Persons Assessed for Cognitive Symptoms” (NorCog) included 18,120 persons referred by general practitioners to outpatient clinics and Norwegian specialist healthcare centres for the assessment of cognitive symptoms and suspected dementia. The data were from the participants’ first visit to the clinics. Because the ability to demonstrate informed consent was an inclusion criterion, only persons living at home with mild and moderate cognitive impairment were included in the registry. Information about falls, medication, and comorbidities was provided by the participant or their next of kin. Voluntary written informed consent was obtained before inclusion. The present study used data from participants with verified dementia diagnoses and information regarding falls in the past 12 months.

### 2.2. Methods

Gait speed, basic functions in activities of daily living (ADL), and the degree of depressive symptoms were the three leading independent frailty-related predictors of falls in our previous study using data from the NorCog [10]. In this follow-up study, we analysed differences in these predictors of falls across the dementia diagnoses and associations between these fall predictors and somatic, mental, and cognitive functions.

### 2.3. Variables

The paper by Medbøen et al. contains the complete list of variables in the registry [11]. In this study, the following variables were used:

#### 2.3.1. Person Characteristics

Sex (male/female); age (years).Physical activity. Easy and strenuous activities were registered, and a physical activity score was calculated as 0–8.Gait speed. The “Short Physical Performance Battery’s” 4-m walking test was used, scoring 0–4 [12,13]. A high score means better performance.Activities of Daily Living (ADL). The “Physical Self Maintenance Scale”, a measure of basic ADL, was used, scoring 6–30. Low scores mean self-reliance [14,15].

#### 2.3.2. Medical History and Cognitive Functions

Depression. Depression in dementia was measured using the Cornell scale, scoring 0–38. Scores ≥12 indicate moderate–severe depression [16]. Cornell measures not only depression but also a range of neuropsychiatric symptoms.Cognitive fluctuations. The MAYO Fluctuation Composite Scale, scoring 0–4, was used to measure cognitive fluctuations [17]. High scores indicate more fluctuations.Cognitive functions. A validated Norwegian version of the Mini-Mental State Examination (MMSE-NR3) screening test was used, scoring 0–30 [18,19]. Higher scores mean better performance.Memory. Delayed recall after 10 min was tested with a demographically adjusted version of the Word List Memory Test (WLMT) in the Consortium to Establish a Registry for Alzheimer’s Disease (CERAD WLMT), scoring 0–10 [20,21,22]. A high score indicates better performance.Drugs. The number of drugs taken regularly was noted and used as a proxy for both comorbidity and side effects of the drugs.

#### 2.3.3. Diagnoses

The diagnoses were discussed in interdisciplinary meetings. All the results of the extensive testing, biological biomarkers (blood and cerebrospinal fluid), MRI or CT scans of the brain, and FDG-PET were evaluated before a conclusion was drawn.

Dementia diagnoses: Dementia was classified as Alzheimer’s disease (AD), mixed AD/vascular dementia (MixADVD), vascular dementia (VD), frontotemporal dementia (FTD), or dementia with Lewy bodies (DLB). The diagnosis of FTD was according to Neary et al. [23], and DLB was diagnosed according to McKeith et al. [24]. The ICD-10 criteria (the 10th revision of the International Statistical Classification of Diseases and Related Health Problems) were used for the other diagnoses. According to ICD-10 F 00.2, the diagnosis of MixADVD was applied to persons with features of both AD and VD.

### 2.4. Statistics

Person characteristics are reported as mean (SD) and number (%). Chi-square and *t*-tests were used to compare fallers and non-fallers. A one-way analysis of variance was used to compare gait speed, ADL, and Cornell scores between the dementia groups, followed by pairwise unadjusted post hoc analyses (least significant differences). The results are shown as mean (SD) and differences between the means (with 95% CI) and *p*-values. Predictors of gait speed, ADL, and Cornell were analysed with a univariate general linear regression model, with age, sex, cognitive functions (MMSE-NR3 and CERAD delayed recall), physical activity, drugs, and dementia diagnoses as independent variables. The results are reported as estimated coefficients (B-values) with 95% CI and *p*-values. Gait speed, ADL, and Cornell for the dementia groups are shown as estimated marginal means with 95% CI. Post hoc comparisons between the dementia groups are reported as differences between the estimated marginal means (least significant difference) with 95% CI and *p*-values. The number of persons in each analysis is shown in the tables. The numbers varied significantly because of missing data. No imputation or other methods for the replacement of missing data were performed. Adding the MAYO score as an independent variable in the multivariable analyses markedly reduced the number of persons in the analyses and the power of the analyses. Therefore, the MAYO score was added as an additional analysis. Statistics were performed with IBM SPSS Statistics for Windows, version 29.0 (IBM Corp., Armonk, NY, USA). A significance level of *p*-values ≤ 0.01 was used to adjust for multiple testing.

## 3. Results

### 3.1. Person Characteristics

The study included 1321 persons with a diagnosis of dementia and information about a fall in the last 12 months. The annual fall incidence was 537/1321 (40.7%). Falls increased with age and with most frailty measures used in this study. The measures of cognition, MMSE-NR3, and CERAD delayed recall did not differ significantly between fallers and non-fallers. Table 1 shows the details and all comparisons between fallers and non-fallers.

### 3.2. Differences in Gait Speed, ADL, and Cornell Scores (The Main Predictors of Falls) Between the Dementia Groups

The three significant and independent predictors of falls in our previous study (gait speed, ADL, and Cornell/depression) differed significantly between the dementia groups. Gait speed and ADL were most impaired in persons with VD and MixADVD and Cornell in persons with DLB. Table 2a presents the results of the three variables divided by the dementia diagnoses and overall statistical comparisons between the diagnoses. Table 2b shows the unadjusted post hoc comparisons of the variables between pairs of diagnoses.

### 3.3. Predictors of Gait Speed, ADL, and Cornell Score

Table 3 shows the predictors of gait speed, ADL, and depression (Cornell). Drugs were associated with all three variables and physical activity with gait speed and ADL. The MAYO score was associated with ADL and Cornell when added as an independent variable. After adjusting for age, sex, cognitive functions (MMSE-NR3 and CERAD delayed recall), physical activity, and the number of drugs, there were still significant differences in ADL between the dementia groups.

## 4. Discussion

Personalised prevention of falls should be directed toward reducing fall-predicting risks in persons with a high incidence of falls. Overall, age and symptoms of frailty were the principal risks of falls in persons with dementia. However, the fall incidence and frailty predictors differed significantly between the dementia diagnoses. In persons with MixADVD and VD, the fall incidence was high, and gait speed and ADL were markedly affected. Also, in persons with DLB, the fall incidence was high but was associated with a high Cornell score, measuring depression and neuropsychological symptoms. In persons with AD and FTD, the fall incidence was low, and gait speed, ADL, and Cornell scores were less affected than in the other groups, indicating less impaired somatic and psychological health as measured with our equipment.

In addition to the significant differences in fall incidence and frailty symptoms between the diagnostic groups (Table 1 and Table 2), the wide standard deviations demonstrated substantial differences within the diagnostic groups. The large interindividual variations likely explain the lack of, or marginal effects of, multifactorial interventions [5,6,7]. As world guidelines propose, preventions tailored to personal impairments are preferred [8,9]. Person-centred prevention of falls necessitates mapping each person’s frailty and other weak functions. This study paid special attention to gait speed, ADL, and Cornell/depression, which were known from our previous study as strong and independent predictors of falls [10]. These variables varied significantly between and within the dementia diagnoses (Table 2a,b). The evidence for a fall risk related to impaired gait speed, ADL (environmental hazards), and depression has been classified as Grades 1B, 1B, and E, respectively, according to a modified GRADE stratification system reported by the world guidelines [8].

In persons with impaired gait speed, often those with VD and MixADVD, prevention should prioritise training to improve gait speed and factors associated with it. Gait speed depends on physical activity (Table 3), specifically physical fitness, endurance, muscular strength, coordination, and balance. Gait speed was also associated with the number of drugs (Table 3), indicating that comorbidities and side effects of medications contribute to reduced gait speed. The fall risk for cardiovascular disease and drugs is considered Grade 1B [8]. Persons with VD and MixADVD have a cerebrovascular disease often complicated with motor and sensory dysfunction and impaired balance and often other diseases like diabetes, cardiovascular disease, and hypertension, all contributing to the fall incidence. In addition, side effects of polypharmacy, such as dizziness, vertigo, somnolence, and hypotension, can enhance unsteadiness and reduce gait speed. Psychoactive drugs, commonly used by persons with dementia, are known to increase the fall risk [25]. Prevention of falls in persons with reduced gait speed demands regular physical training, optimal treatment of comorbidities, and a review and revision of drug therapy.

Basic ADL measures several practical issues at home, including personal hygiene and toilet functions, assistance with eating and dressing, and functional mobility indoors and outdoors. Therefore, improving ADL (a Grade 1B risk factor) necessitates various interventions. These complex functions were markedly reduced in persons with VD and MixADVD, probably due to widespread cerebral lesions. Impaired ADL was associated with reduced physical activity, drugs (i.e., comorbidities and side effects of drugs), reduced cognitive functions (MMSE-NR3), and cognitive fluctuations (MAYO score). Physical reduction and comorbidity reduce practical performance, and cognitive impairment and side effects of drugs affect the ability to plan and carry out complex functions such as ADL. Cognitive fluctuations with disturbed alertness hinder long-lasting mental activity, as shown by the association between ADL and the MAYO score. Improving ADL includes physical training, simplification and training of daily activities, optimal treatment of comorbidities, revision of pharmacotherapy, and treatment of neuropsychiatric symptoms. After adjusting for all variables, significant differences in ADL remain between the diagnostic groups.

Cornell, a measure of depression, was highly statistically significant and independently predicted falls in our previous study. Cornell scores differ from gait speed and ADL in that they are highly influential in persons with DLB. Depression is considered Grade E in the world guidelines [8]. Cornell measures not only depression, which could explain the discrepancy between our results and the guidelines, but is also a composite score of neuropsychological symptoms: anxiety, irritability, changed behaviour, restlessness, somatic symptoms (weight loss and abdominal complaints), sleep disturbances, delusions, hallucinations, and suicide intention. These symptoms are typical of persons with DLB [26,27,28]. The associations between MAYO, a measure of cognitive fluctuations and sleep disturbances, on one side and Cornell (a neuropsychiatric composite score) and DLB were expected. Cognitive fluctuation occurs in up to 90% of persons with DLB and is used as a diagnostic criterion, but it appears in all types of dementia [29]. The pathophysiology of cognitive fluctuations is unknown. Prevention of falls in persons with high Cornell scores (depression and neuropsychiatric symptoms) and cognitive fluctuations/sleep disturbances is psychological and symptomatic treatment with drugs.

Increasing age, but not sex, was associated with reduced gait speed and falls. Reduced physical performance, comorbidity, and polypharmacy seemed to be the most important triggers of falls, as they affect both gait speed and ADL. The effects of comorbidity (primarily somatic but also psychiatric comorbidity) and polypharmacy deserve special attention, since they affect gait speed, ADL, and Cornell. Psychotropic medication, often used by persons with cognitive impairment, increases the rate of falls, but anti-dementia drugs (cholinesterase inhibitors) do not [25,30].

Dementia and cognitive impairment increase the risk of falls significantly, and assessment of cognition as a risk of falls has received a Grade 1B recommendation [8,31]. Neither in this study nor in our previous one was the degree of cognitive impairment, measured as MMSE-NR3 and recall (CERAD delayed recall), an independent predictor of falls, except for the association between MMSE-NR3 and ADL. The exclusion of persons with severe cognitive impairment might explain the lack of association in this study. Trail-Making Test B (TMT-B), a measure of set-shifting executive functions, has been judged to predict falls in other studies [2]. NorCog included this test, but only a minority of persons with dementia had completed the test, and all results were above the upper limit. Our findings indicate that falls in persons with dementia are related to somatic disorders, neuropsychiatric symptoms, and drug treatment and not specifically to cognitive decline.

### Strengths and Limitations

The NorCog included consecutive persons attending 45 outpatient clinics from all over Norway, assuring the representativeness of the participants. Data were from the first clinic visit in the early stage of the disease with mild to moderate cognitive impairment, which probably influenced the results. Validated methods were used for the numerous and distinct registrations of somatic, psychiatric, and cognitive functions; psychosocial factors; diseases; blood and spinal fluid tests; and cerebral images. The final diagnoses of dementia were a consensus in multidisciplinary meetings based on all available information and internationally accepted diagnostic criteria.

Information about falls, medication use, and comorbidities relies on information provided by the person or their next of kin rather than from health personnel or validated instruments and is inaccurate. To avoid uncertainty about the precise type of comorbidity and drugs, the number of regular intakes of drugs was used as a proxy for comorbidities and side effects of medications. This shortcoming made the associations between falls and comorbidities and drugs defective. No information on nutritional status, a significant predictor of falls, was available. The limited number of persons with FTD makes the results in this group less reliable. Another limitation was the relatively high proportion of missing data, resulting in a reduced and variable number of persons in the analyses. Therefore, the number of subjects in each analysis is shown in the tables. In searching for personalised predictors of falls, the starting point was the results of our previous study from the same registry. The previous study showed that gait speed, ADL, and Cornell were the main and independent predictors of falls. This premise might have introduced bias. To reduce type I errors in this explorative study with multiple testing and unadjusted post hoc analyses, *p* ≤ 0.01 was judged statistically significant. Despite the adjustment of the significant value, neither type I nor type II errors could be excluded. Statistically significant associations in cross-sectional studies do not prove causal relations.

## 5. Conclusions

The study highlights that preventing falls in persons with dementia prone to such incidents requires personalised interventions. Personalised fall prevention should explore each individual’s independent fall-predicting functions (gait speed, ADL, and Cornell/depression) and focus interventions on the weak functions in persons with a high fall incidence. In persons with VD and MixADVD (a group with a high incidence of falls) and reduced gait speed, interventions should focus on improving physical fitness, endurance, muscular strength, coordination, and balance, as well as optimising treatment of comorbidities and medication use. Improving impaired ADL, often in persons with VD and MixADVD, necessitates the training of complex behavioural, neuropsychological, and physical functions, in addition to optimising treatment of comorbidities and medications. In persons with DLB, depression, and fluctuations (a group with a high fall incidence), psychological and pharmacological interventions are preferred. Personalised fall prevention is demanding but appears to be the most effective way to reduce falls and fall-related injuries in persons with cognitive impairment.

## Figures and Tables

**Table 1 geriatrics-10-00106-t001:** Person characteristics, dementia diagnoses, and frailty variables are divided into persons with and without a fall. The results are shown as numbers with a proportion (%) and mean with SD. Chi-square and *t*-tests were used to compare fallers and non-fallers.

Variables	Number	Fall	Not Fall	Statistics ^1^
Sex—Female —Male	706 615	285 (40%) 252 (41%)	421 (60%) 363 (59%)	*p* = 0.866
Age	1321	78.4 (7.4)	75.9 (7.6)	***p* < 0.001**
**Dementia diagnoses**				
AD	742	240 (32%)	502 (68%)	***p* < 0.001**
MixADVD	249	126 (51%)	123 (49%)	
VD	208	115 (55%)	93 (45%)	
FTD	42	12 (29%)	30 (71%)	
DLB	80	44 (55%)	36 (45%)	
**Frailty variables**				
Gait speed	570	3.01 (1.04)	3.36 (0.85)	***p* < 0.001**
ADL	1297	9.72 (3.80)	7.64 (2.56)	***p* < 0.001**
Cornell	969	7.55 (5.52)	6.01 (5.06)	***p* < 0.001**
MMSE-NR3	1283	20.57 (4.37)	20.68 (4.53)	*p* = 0.673
CERAD delayed recall	1055	1.39 (1.74)	1.23 (4.01)	*p* = 0.423
Physical activity	1152	2.53 (2.18)	3.45 (2.39)	***p* < 0.001**
Drugs (number/day)	1004	5.14 (3.29)	3.81 (2.91)	***p* < 0.001**
MAYO score	421	1.84 (1.32)	1.42 (1.20)	***p* < 0.001**

^1^ Significant *p*-values in bold font.

**Table 2 geriatrics-10-00106-t002:** (**a**). The gait speed, ADL, and Cornell scores divided by the dementia diagnoses. The results are shown as numbers and mean (SD). Statistics are the overall comparisons between the diagnostic groups. (**b**). Post hoc pairwise differences in gait speed, ADL, and Cornell scores between the dementia groups. The results are presented as differences between the means with 95% confidence intervals and significance values (*p*-values).

a							
Variables	No	AD	MixADVD	VD	FTD	DLB	Statistics ^1^
Gait speed	577	3.40 (0.87)	3.06 (0.99)	2.81 (0.92)	3.33 (0.89)	3.29 (1.05)	***p* < 0.001**
ADL	1315	7.66 (2.59)	9.34 (3.56)	10.39 (4.09)	8.74 (4.37)	8.53 (2.75)	***p* < 0.001**
Cornell	981	6.25 (5.07)	6.71 (5.58)	6.77 (5.50)	8.26 (5.98)	9.27 (5.87)	***p* < 0.001**
**b**							
**Comparisons Between** **Dementia Groups**		**Gait Speed**		**ADL**		**Cornell**	
AD vs. MixADVD		0.34 (0.15; 0.54)		−1.69 (−2.13; −1.24)		−0.46 (−1.35; 0.42)	
		***p* < 0.001**		***p* < 0.001**		*p* = 0.306	
AD vs. VD		0.60 (0.39; 0.81)		−2.73 (−3.21; −2.25)		−0.52 (−1.49; 0.44)	
		***p* < 0.001**		***p* < 0.001**		*p* = 0.286	
AD vs. FTD		0.07 (−0.46; 0.60)		−1.08 (− 2.05; −0.11)		−2.01 (−3.94; −0.08)	
		*p* = 0.790		*p* = 0.029		*p* = 0.041	
AD vs. DLB		0.12 (−0.20; 0.44)		−0.87 (−1.61; −0.13)		−3.02 (−4.45; −1.59)	
		*p* = 0.464		*p* = 0.021		***p* < 0.001**	
MixADVD vs. VD		0.25 (0.00; 0.50)		−1-05 (−1.62; −0.48)		−0.06 (−1.21; 1.09)	
		*p* = 0.048		***p* < 0.001**		*p* = 0.915	
MixADVD vs. FTD		−0.27 (−0.82; 0.27)		−0.61 (−0.42; 1.62)		−1.55 (−3.57; 0.48)	
		*p* = 0.326		*p* = 0.245		*p* = 0.134	
MixADVD vs. DLB		−0.23 (−0.57; 0.12)		0.82 (0.02; 1.62)		−2.56 (−4.12; −1.00)	
		*p* = 0.203		*p* = 0.046		***p* = 0.001**	
VD vs. FTD		−0.53 (−1.09; 0.03)		1.65 (0.62; 2.69)		−1.49 (−3.55; 0.57)	
		*p* = 0.062		***p* = 0.002**		*p* = 0.157	
VD vs. DLB		−0.48 (−0.83; −0.12)		1.87 (1.05; 2.69)		−2.50 (−4.10; −0.89)	
		*p* = 0.009		***p* < 0.001**		***p* = 0.002**	
FTD vs. DLB		0.05 (−0.55; 0.65)		0.21 (−0.96; 1.39)		−1.01 (−3.33; 1.30)	
		*p* = 0.876		*p* = 0.724		*p* = 0.391	

^1^ Significant *p*-values in bold font.

**Table 3 geriatrics-10-00106-t003:** Predictors of gait speed, ADL, and Cornell scores analysed with a univariate general linear regression model ^1^.

Independent Variables	Dependent Variable Gait Speed (No. 369)	Dependent Variable ADL (No. 723)	Dependent Variable Cornell (No. 555)
Sex (male)	0.202 (0.024; 0.380)	0.277 (−0.715; 0.161)	−0.067 (−0.887; 1.021)
	*p* = 0.026	*p* = 0.215	*p* = 0.891
Age (years)	−0.022 (−0.036; −0.007)	0.030 (−0.002; 0.062)	0.031 (−0.039; 0.100)
	***p* = 0.004**	*p* = 0.066	*p* = 0.390
MMSE-NR3	0.015 (−0.007; 0.037)	−0.134 (−0.190; −0.078)	−0.085 (−0.210; 0.040)
	*p* = 0.169	***p* < 0.001**	*p* = 0.183
CERAD delayed recall	0.023 (−0.038; 0.084)	0.140 (−0.011; 0.291)	0.233 (−0.097; 0.563)
	*p* = 0.461	*p* = 0.070	*p* = 0.166
Physical activity	0.060 (0.018; 0.103)	−0.290 (−0.391; −0.190)	−0.084 (−0.301; 0.133)
	***p* = 0.005**	***p* < 0.001**	*p* = 0.447
Drugs	−0.044 (−0.075; −0.013)	0.230 (0.157; 0.304)	0.259 (0.102; 0.415)
(number/day)	***p* = 0.005**	***p* < 0.001**	***p* = 0.001**
Dementia diagnoses	*p* = 0.048	***p* < 0.001**	*p* = 0.022
MAYO (addition)	−0.187 (−0.337; −0.037)	0.631 (0.304; 0.959)	2.010 (1.409; 2.612)
	(no 116) *p* = 0.015	(no 242) ***p* < 0.001**	(no 220) ***p* < 0.001**
**Dementia groups**			
AD	3.414 (3.292; 3.537)	7.920 (7.626; 8.214)	6.622 (5.985; 7.260)
MixADVD	3.339; 3.129; 3.549)	8.700 (8.198; 9.203)	6.556 (5.435; 7.677)
VD	3.025 (2.814; 3.235)	9.665 (9.120; 10.211)	6.323 (5.141; 7.505)
FTD	3.156 (2.463; 3.848)	10.041 (8.681; 11.401)	7.084 (4.200; 9.969)
DLB	3.291 (2.927; 3.654)	8.156 (7.175; 9.138)	10.222 (8.135; 12.309)
**Post hoc** **comparisons**			
AD vs. MixADVD	0.075 (−0.174; 0.325)	0.780 (−1.374; −0.187)	0.066 (−1.247; 1.380)
	*p* = 0.553	** *p* ** **= 0.010**	*p* = 0.921
AD vs. VD	0.390 (0.139; 0.641)	−1.745 (−2.385; −1.106)	0.300 (−1.093; 1.692)
	***p* = 0.002**	***p* < 0.001**	*p* = 0.673
AD vs. FTD	0.259 (−0.446; 0.963)	−2.121 (−3.515; −0.727)	−0.462 (−3.422; 2.498)
	*p* = 0.470	***p* = 0.003**	*p* = 0.759
AD vs. DLB	0.124 (−0.261; 0.509)	−0.236 (−1.268; 0.795)	−3.600 (−5.797; −1.403)
	*p* = 0.527	*p* = 0.653	***p* = 0.001**
MixADVD vs. VD	0.314 (0.025; 0.604)	−0.965 (−1.691; −0.240)	0.233 (−1.361; 1.827)
	*p* = 0.033	***p* = 0.009**	*p* = 0.774
MixADVD vs. FTD	0.183 (−0.543; 0.909)	1.340 (−2.810; 0.129)	−0.528 (−3.659; 2.603)
	*p* = 0.620	*p* = 0.074	*p* = 0.740
MixADVD vs. DLB	0.048 (−0.372; 0.468)	0.544 (−0.561; 1.648)	−3.666 (−6.044; −1.288)
	*p* = 0.841	*p* = 0.334	***p* = 0.003**
VD vs. FrTeD	−0.131 (−0.853; 0.591)	−0.375 (−1.838; 1.088)	−0.761 (−3.871; 2.348)
	*p* = 0.721	*p* = 0.615	*p* = 0.631
VD vs. DLB	0.266 (−0.685; 0.153)	1.509 (0.393; 2.625)	−3.899 (−6.276; −1.523)
	*p* = 0.213	***p* = 0.008**	***p* = 0.001**
FTD vs. DLB	−0.135 (−0.914; 0.644)	1.884 (0.236; 3.533)	−3.138 (−6.638; 0.362)
	*p* = 0.734	*p* = 0.025	*p* = 0.079

^1^ The results of the regression model are shown as estimated coefficients (B-values) with 95% CI and *p*-values. The levels of gait speed, ADL, and Cornell in the dementia groups are shown as estimated marginal means with 95% CI. The post hoc pairwise comparisons are shown as differences between the estimated marginal means with 95% CI and *p*-values. Significant *p*-values are in bold font.

## Data Availability

The national registry NorCog is responsible for the source data. The deidentified data files from persons with falls were transferred to Innlandet Hospital Trust Brumunddal, Norway, and stored on a server dedicated to research. The security follows the rules given by The Norwegian Data Protection Authority, P.O. Box 8177 Dep. NO-0034 Oslo, Norway. The data are available upon request to the author.

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
