# Peer review of "Personalised Prevention of Falls in Persons with Dementia—A Registry-Based Study"

_geriatrics, 2025, doi:10.3390/geriatrics10040106_

Round 1

Reviewer 1 Report

Comments and Suggestions for Authors
The manuscript entitled “Prevention of Falls in Persons with Dementia Based on Two Personalized Risk Factors – A Registry-Based Study” addresses the identification of predictors of falls related to frailty in individuals with dementia, through the analysis of longitudinal data recorded in the Norwegian Registry of Persons Assessed for Cognitive Symptoms (NorCog). The article seeks to explore differences between dementia diagnoses and changes in gait speed, depressive symptoms, and basic activities of daily living, aiming to fill gaps in the literature. As points for minor revision, I suggest: reviewing the definition of the term "fall" addressed in the work (was it based on reports from the caregiver, the physician, or a validated instrument?); citing or evaluating other relevant variables, such as medication use, comorbidities, and the criteria used to classify dementia subtypes. Another point that could enrich the study would be to highlight its implications for public and/or clinical policies. The study could become more compelling with a more in-depth discussion and clearer detailing of the methods.
Comments on the Quality of English Language
 The manuscript would benefit from minor grammatical and syntactical revisions to improve clarity and fluency.

Author Response

Response to Reviewer 1

The manuscript entitled “Prevention of Falls in Persons with Dementia Based on Personalized Risk Factors – A Registry-Based Study” addresses the identification of predictors of falls related to frailty in individuals with dementia, through the analysis of longitudinal data recorded in the Norwegian Registry of Persons Assessed for Cognitive Symptoms (NorCog). The article seeks to explore differences between dementia diagnoses and changes in gait speed, depressive symptoms, and basic activities of daily living, aiming to fill gaps in the literature.

As points for minor revision, I suggest: reviewing the definition of the term "fall" addressed in the work (was it based on reports from the caregiver, the physician, or a validated instrument?); citing or evaluating other relevant variables, such as medication use, comorbidities, and the criteria used to classify dementia subtypes.

Response:

Information about falls, medication use, and comorbidities relies on information provided by the person or their next of kin, rather than from health personnel or validated instruments. This information has been added as a limitation on page 10, lines 295-297.

The criteria used to classify dementia subtypes are provided on page 3-4, lines 123-129, with references to the relevant criteria. No changes have been made.

Comment:

Another point that could enrich the study would be to highlight its implications for public and/or clinical policies. The study could become more compelling with a more in-depth discussion and clearer detailing of the methods.

Response:

To highlight the implications, one sentence has been added at the end of the paper (page 10, lines 328-330). Detailing the methods for all variations in personalised prevention requires a significant enlargement of the paper and has not been performed.

Comment: The manuscript would benefit from minor grammatical and syntactical revisions to improve clarity and fluency.

Response:

English is not my native language. I have written the text and then used the application “Grammarly” to improve it. I have now reworked the paper and made several minor changes. Together with the additional changes that will probably be made by “Geriatrics” during the publication process, I hope it will be satisfactory.

Reviewer 2 Report

Comments and Suggestions for Authors

This manuscript addresses a crucial topic in geriatric care: fall prevention in individuals with dementia. The authors leverage a well-established national registry (NorCog) and build upon their previous work to explore fall risks across dementia subtypes. The analysis is statistically robust, clinically relevant, and written clearly. However, several methodological, interpretative, and editorial aspects should be addressed before acceptance.

  • The focus on dementia subtypes (AD, VD, MixADVD, DLB, FTD) in relation to specific fall predictors (gait speed, ADL, depression) is both timely and relevant. The personalization approach adds value. However, the novelty is modest, as the study largely reuses the same registry and key variables from the authors’ prior publication (Ref 10). Please clarify how this study extends previous findings beyond subgroup analysis.
  • The use of a cross-sectional design limits causal inferences. Please discuss this limitation more clearly.
  • Missing data is a significant concern. Some analyses include fewer than 400 of the 1321 participants. The authors acknowledge this but should quantify the missingness per variable and describe handling methods (e.g., listwise deletion, imputation).
  • A p-value threshold of 0.01 was used, but no correction for multiple comparisons was performed. Please justify this more clearly and discuss potential Type I error inflation.
  • The use of univariate linear regression for Table 3 is potentially misleading. Multivariate models adjusting for all covariates would be more appropriate for identifying independent predictors. Clarify whether the models are truly univariate or multivariate.
  • The interpretation of Cornell scores as indicators of depression may be overstated. As acknowledged, the scale includes a wide range of neuropsychiatric symptoms. This conflation may mislead clinicians. Please distinguish “depression” from broader neuropsychiatric burden more consistently.
  • Line 18: Please define MixADVD first mention.
  • Line 23: “Annual fall incidence” could be misinterpreted as a population-level rate. Consider rephrasing: “Among study participants, 40.7% reported a fall in the past year.”
  • Line 77: Please provide exact inclusion/exclusion criteria, particularly regarding dementia diagnosis and fall reporting.
  • Line 86–87: You reference Medbøen et al. for variables. A brief table summarizing key variables with definitions would enhance clarity.
  • Line 120: Reference to a previous paper for variable definitions is insufficient. Definitions should be self-contained in the Methods section.
  • The manuscript is generally well written, but several long sentences could be shortened for clarity.
  • Replace instances of informal language such as “no or only marginal effects” (line 13) with “minimal or non-significant effects.”
  • Please proofread carefully for minor grammatical errors and inconsistent formatting.

Author Response

Response to Reviewer 2

Comment:

The English could be improved to more clearly express the research.

Response:

The same answer as given to Reviewer 1. English is not my native language. I have written the text and then used the application “Grammarly” to improve it. I have now reworked the paper, making several minor changes. Together with the additional changes that will probably be made by “Geriatrics” during the publication process, I hope it will be satisfactory.

Comment:

This manuscript addresses a crucial topic in geriatric care: fall prevention in individuals with dementia. The authors leverage a well-established national registry (NorCog) and build upon their previous work to explore fall risks across dementia subtypes. The analysis is statistically robust, clinically relevant, and written clearly. However, several methodological, interpretative, and editorial aspects should be addressed before acceptance.

The focus on dementia subtypes (AD, VD, MixADVD, DLB, FTD) in relation to specific fall predictors (gait speed, ADL, depression) is both timely and relevant. The personalization approach adds value. However, the novelty is modest, as the study largely reuses the same registry and key variables from the authors’ prior publication (Ref 10). Please clarify how this study extends previous findings beyond subgroup analysis.

Response:

Our previous study (Ref 10) showed that reduced gait speed, impaired basic ADL, and depression were independent and the main predictors of falls. This information alone is not sufficient for personalised prevention of falls. In this follow-up study, we analysed differences in these predictors of falls across the dementia diagnoses (Table 2), and associations between these fall predictors and somatic, mental, and cognitive functions available for personalised prevention (Table 3). This information allows targeted, personalised interventions. This study is a logical follow-up to our previous study, without any overlap between the two studies.

 Comment:

The use of a cross-sectional design limits causal inferences. Please discuss this limitation more clearly.

Response:

It is well known that statistically significant associations in a cross-sectional study do not prove causal relations.  One sentence has been added, page 10, lines 311-312.

Comment:

Missing data is a significant concern. Some analyses include fewer than 400 of the 1321 participants. The authors acknowledge this but should quantify the missingness per variable and describe handling methods (e.g., listwise deletion, imputation).

Response:

Since the total number of participants in the study and the number in each analysis are provided, further quantification of the missingness per variable seems unnecessary. No imputation or other methods for replacement of missing data were performed. This has been added, page 4, lines 145-146.  

Comment:

A p-value threshold of 0.01 was used, but no correction for multiple comparisons was performed. Please justify this more clearly and discuss potential Type I error inflation.

Response:

Choosing a p-value of 0.01 instead of 0.05 is a pragmatic way to adjust for multiple comparisons. But independent of the type of adjustment, neither Type I nor Type II errors can be excluded. One sentence has been added, page 10, lines 310-311.

Comment:

The use of univariate linear regression for Table 3 is potentially misleading. Multivariate models adjusting for all covariates would be more appropriate for identifying independent predictors. Clarify whether the models are truly univariate or multivariate.

Response:

The definition of words might be confusing. Univariate means one dependent variable, multivariate means two or more dependent variables.  Multivariable means two or more independent predictors (covariates). In this study, the regression analyses are univariate multivariable analyses. No changes have been made.

Comment:

The interpretation of Cornell scores as indicators of depression may be overstated. As acknowledged, the scale includes a wide range of neuropsychiatric symptoms. This conflation may mislead clinicians. Please distinguish “depression” from broader neuropsychiatric burden more consistently.

Response:

The information about the Cornell score has been added to page 3, lines 106-107, and the difference between the Cornell score and depression is mentioned several times in the text.  

Comment:

Line 18: Please define MixADVD first mention.

Response:

We have written in the paper on page 3, lines 124-126: “The ICD-10 criteria (the 10th revision of the International Statistical Classification of Diseases and Related Health Problems) were used for the other diagnoses”. Thus, the ICD-10 criteria also apply to MixADVD. MixADVD is defined in ICD-10 F 00.2 as “ Dementia in Alzheimer disease, atypical or mixed type”. We have added the following information on page 4, lines 128-129: “According to ICD-10 F 00.2, the diagnosis of MixADVD was given to persons with features of both AD and VD”.

Comment:

Line 23: “Annual fall incidence” could be misinterpreted as a population-level rate. Consider rephrasing: “Among study participants, 40.7% reported a fall in the past year.”

Response:

The change has been made, page 1. Lines 24-25.

Comment:

Line 77: Please provide exact inclusion/exclusion criteria, particularly regarding dementia diagnosis and fall reporting.

Response:

A more detailed description of the inclusion/exclusion criteria is given on page 2, lines 77-78, and lines 80-81.

Comment:

Line 86–87: You reference Medbøen et al. for variables. A brief table summarizing key variables with definitions would enhance clarity.

Response:

To give information on all variables in the NorCog registry in addition to the point-by-point description and definitions of the variables used in this study (pages 3 and 4), seems unnecessary.

Comment:

Line 120: Reference to a previous paper for variable definitions is insufficient. Definitions should be self-contained in the Methods section.

Response:

Line 120 (line 130 in the revised manuscript) has been deleted. The variable definitions are sufficiently described, and the differences between the wording in this paper and our previous one are unimportant.  

Comment:

The manuscript is generally well written, but several long sentences could be shortened for clarity.

Response:

I refer to my answer to the reviewer’s first comment.  I have worked through the paper and made several improvements.

Comment:

Replace instances of informal language such as “no or only marginal effects” (line 13) with “minimal or non-significant effects.”

Response:

The wording has been changed (page 1, line 14)

Comment:

Please proofread carefully for minor grammatical errors and inconsistent formatting.

Response:

Several minor changes have been made!

Reviewer 3 Report

Comments and Suggestions for Authors

Please see doc attachement.

Author Response

Response to Reviewer 3

Comment:

Quality of English Language:  The English is fine and does not require any improvement.

Response:

Thanks!

Comment:

Title of the article:

Prevention of Falls in Persons with Dementia based on Personalised Risk Factors — A Registry-Based Study

Strengths:

The title is clear and directly reflects the study’s focus. But can be improved.

Suggested Revision:

(1) —The Prevention of Falls in Persons with Dementia based on Personalised Risk Factors: A Registry-Based Study

(2) - Fall Prevention in Dementia: A Personalised Registry-Based Study

Response:

The title has been changed to a third alternative:

“Personalised Prevention of Falls in Persons with Dementia – A Registry-Based Study”

Resume and review (what the paper is about):

This study explored person with falls risk related to physical, mental, cognitive functions and types of dementia such as Alzheimer’s, Vascular, and Mixed Alzheimer’s Disease vs Vascular Dementia, Frontotemporal and Dementia with Lewy Bodies. The age and symptoms of frailty were the principal risks of falls in people with dementia, although the diagnoses of dementia differed. With this study the author’s hypothesized that personalized prevention is recommended for each individual’s fall risk factors.

Key strengths (of this paper):

Present a very good insights to the existing literature.

Specific comments:

Abstract:

Strengths: The abstract answer the tree major questions: Why, How and What! Well, done!

Area for improvement: Keywords.

  1. a) Keywords refinement: The words included in the title are not necessary mention

again (like: Dementia). Please delete.

  1. b) Keywords should be written in small caps. Please correct.

Response:

The changes to the keywords have been performed (page 1, lines 36-37).

  1. Introduction:

General background: Sound but need to be improved.

Suggested Revision: It is necessary to include a sentence/s about Alzheimer disease, Vascular dementia, Frontotemporal dementia and Dementia with Lewy Bodies (the extent to which these types of dementia affect physical, mental and cognitive functions).

Response:

A detailed description of physical, mental and cognitive functions in the four types of dementia seems unnecessary. One general sentence covering the topic has been added (page 2, lines 43-44).

Comment:

  1. Materials and Methods:

Strengths: Key chapter summarized.

Area for improvement: Need a sub-section Sample (my own opinion).

Suggestion revision: 2.2 Sample. The sample consisted of 1321 composed by Female (n = 706) and Male (n = 615). In the whole sample of persons, Sfall: Female (n = 285 (40%)), Male (n = 252 (41%)); Snotfall: Female (n = 421 (60%)), Male 363 (59%)); Sage = 1321 (?) (Fall (78.4 + 7.4), Not Fall (75.9 + 7.6) with statistically significant (p < 0.001).

Pay attention: If the authors agreed with this chance, please remove from the Table 1. No need to repeat data.

Response:

I prefer not to perform the proposed changes. The description of the sample (number of persons, sex, falls, etc., with comparisons between the groups) is a result of the inclusion/exclusion criteria and is part of section 3. Results. I prefer to present the results in the table rather than in the text because readability is generally better in a table – I think.

Comment:

  1. Results:

Strengths: Statistical analysis.

Areas for improvement: Data of no interest to the study referred to in the Tables 1 and 3.

Suggestion revision: Delete entire line of the variable sex (female and male — Table 1) and sex (male — Table 3). Are not relevant data for the study. Not least because the data (table 3) does nothing to support the results of the study. My question is: Why the authors assess the differences between female and male? This is not in line with any aim of the study, nor was it covered in the introduction chapter.

Response:

It is correct that sex and differences between males and females were not the aim of the study. But the sex perspective, including differences between males and females, is nearly obligatory in all clinical research. It is also correct that the sex perspective does not support the results. However, it is nice to know that there were no significant differences between males and females. No changes have been made.

Comment:

4, Discussion:

General background: Sound but need to be improved.

Strengths: Unexpected results (but which are not discussed and explained).

Areas for improvement (1): A deep discuss is needed with the unexpected results (Table 3 - dementia groups and Post hoc comparisons results).

Areas for improvement (2): A deep discuss is needed with the assumption of the study aim. Please create subsections for Alzheimer dementia, Vascular dementia, Mixed Alzheimer vs Vascular dementia, Frontotemporal dementia, and Dementia with Lewy Bodies to physical, mental and cognitive functions.

Response:

The reviewer mentions unexpected results. None of the results were unexpected and have not been described as unexpected. The differences between the dementia groups are important new knowledge. These differences have been mentioned several times in the discussion. The differences must be considered when planning personalised fall prevention. The discussion is structured in paragraphs based on the three important predictors of falls: gait speed, ADL, and Cornell/depression. To restructure the discussion completely and divide it into new paragraphs/subsections for Alzheimer dementia, Vascular dementia, Mixed Alzheimer/Vascular dementia, Frontotemporal dementia, and Dementia with Lewy Bodies, and physical, mental, and cognitive functions will deviate from the structure of the paper. No changes have been made.  

Comment:

  1. Conclusions:

Strengths: Summarizes key findings.

Flaws: No stronger take-home message.

Areas for Improvement: The conclusions should directly answer the research objectives and hypotheses.

Response:

The study is exploratory, not hypothesis-driven, and does not involve testing a single hypothesis that is accepted or rejected.  Therefore, a short take-home message is not appropriate, and a little longer summary of the results is preferable, such as the “Conclusions”.

Comment:

Example addition: “The study highlights that preventing falls in person prone to such incidents requires individual prevention measures. By evaluating factors like walking speed, daily activities, and depression, the study concludes that a focus on functional interventions is essential. The professional technicians should aim to identify and implement appropriate methods, suitable means and solutions for those with a high risk of falling.” (Lines 293-295).

Lines 295-301: “In persons with ... comorbidity and medication”. This sentence needs to be improved. Please do it.

Lines 301-302: The study deals with prevention techniques, not psychological or pharmacological intervention. Please delete, it doesn’t make sense.

Response:

Parts of the reviewer’s example have been added to the first part of the conclusion (lines 314-315). Lines 295-301 (lines in the revised manuscript: 321-323) have been revised somewhat. Lines 301-302 (lines in the revised manuscript: 326-328) have not been deleted because psychological and pharmacological interventions are the preferred methods for preventing falls in persons with DLB, depression, and fluctuations, as shown in Table 3.

  1. References

Strengths: The references list is comprehensive and relevant.

Comment:

  1. Potential contribution and impact to the field:

With this minor improve, this paper has big potential to follow for publication.

Response:

Thank you – we look forward to the publication of the revised paper.

Round 2

Reviewer 2 Report

Comments and Suggestions for Authors

I appreciate the authors' efforts in addressing all the comments and suggestions provided. The manuscript is now clearer and scientifically sound. The revisions have improved the overall quality of the study, and the data support the conclusions effectively.